# Transcriptomic Analysis of Genes Involved in Plant Defense Response to the Cucumber Green Mottle Mosaic Virus Infection

**DOI:** 10.3390/life11101064

**Published:** 2021-10-10

**Authors:** Anna Slavokhotova, Tatyana Korostyleva, Andrey Shelenkov, Vitalii Pukhalskiy, Irina Korottseva, Marina Slezina, Ekaterina Istomina, Tatyana Odintsova

**Affiliations:** 1Chumakov Federal Scientific Center for Research and Development of Immune-and-Biological Products of Russian Academy of Sciences, 108819 Moscow, Russia; annslav82@gmail.com; 2Vavilov Institute of General Genetics, Russian Academy of Sciences, 119991 Moscow, Russia; tatkor@vigg.ru (T.K.); pukhalsk@vigg.ru (V.P.); omey@list.ru (M.S.); mer06@yandex.ru (E.I.); odintsova2005@rambler.ru (T.O.); 3Central Research Institute of Epidemiology, Rospotrebnadzor, 111123 Moscow, Russia; 4Federal Scientific Vegetable Center, 143072 Moscow, Russia; korottseva@mail.ru

**Keywords:** CGMMV, RNA-seq, plant bioinformatics, disease resistance, stress tolerance, cucumber, plant-virus interaction

## Abstract

Plants have evolved a complex multilayered defense system to counteract various invading pathogens during their life cycle. In addition to silencing, considered to be a major molecular defense response against viruses, different signaling pathways activated by phytohormones trigger the expression of secondary metabolites and proteins preventing virus entry and propagation. In this study, we explored the response of cucumber plants to one of the global pathogens, cucumber green mottle mosaic virus (CGMMV), which causes severe symptoms on leaves and fruits. The inbred line of *Cucumis sativus* L., which is highly susceptible to CGMMV, was chosen for inoculation. Transcriptomes of infected plants at the early and late stages of infection were analyzed in comparison with the corresponding transcriptomes of healthy plants using RNA-seq. The changes in the signaling pathways of ethylene and salicylic and jasmonic acids, as well as the differences in silencing response and expression of pathogenesis-related proteins and transcription factors, were revealed. The results show that silencing was strongly suppressed in infected plants, while the salicylic acid and ethylene signaling pathways were induced. The genes encoding pathogenesis-related proteins and the genes involved in the jasmonic acid pathway changed their expression insignificantly. It was also found that WRKY and NAC were the most sensitive to CGMMV infection among the transcription factors detected.

## 1. Introduction

During their life cycle, plants are exposed to various invaders, including fungi, bacteria, viruses, nematodes, and pests. In order to survive and prevent pathogen entrance and propagation, plants have evolved a complex multilayered defense system comprising a set of structural and chemical barriers. The first line of defense includes structural shields such as cuticular wax, rigid lignin depositions on the cell walls, and specific stomatal structure with protected apertures [1]. The second line of defense is trigged by pathogen-associated molecular patterns (PAMPs) that represent different microbial structures, such as flagellin, lipopolysaccharides, and fungal cell wall components [2]. These invader’s components are recognized by specific plant receptors and induce pattern-triggered immunity (PTI) [3]. Plant pathogens, in turn, have developed mechanisms to overcome PTI by secreting effector proteins into the plant cells, which manipulate host factors and suppress the plant defenses.

Pathogen invasion triggers numerous plant signaling pathways that induce the biosynthesis of secondary metabolites and the expression of pathogenesis-related (PR) and other defense proteins with antimicrobial properties. Signaling pathways are activated by phytohormones whose sophisticated crosstalk facilitates balanced response to changeable environmental conditions. Three major phytohormones considered to play a pivotal role in biotic stress resistance are salicylic acid (SA), ethylene (ET), and jasmonic acid (JA) [4]. The SA pathway is mainly induced by biotrophic pathogens, while JA and ET activate response against necrotrophic pathogens and insects [5]. Furthermore, phytohormones modulate the expression of different transcription factors (TFs) responsible for rapid transcriptional reprogramming upon pathogen entry [4].

PTI is regarded as a major defense mechanism against non-viral pathogens, while the primary response to viral invasion is virus-induced gene silencing (VIGS), an ancient mechanism that directly defends host cells against foreign nucleic acids, including viruses and active transposable elements [6]. Meanwhile, recent studies of plant immunity have revealed that phytohormone-dependent signaling pathways are also activated in response to viral infection [7].

CGMMV is a member of the *Tobamovirus* genus that causes diseases of Cucurbitaceae plants: cucumber, pumpkin, watermelon, melon, squash, gourds, etc. [8,9]. The infected plants exhibit stunted growth, fruit distortion, mottling, and mosaic on leaves. Outbreaks of the disease sporadically occur in the field; however, they are more common in greenhouses, causing up to 40% yield loss [9]. CGMMV was first discovered in England in 1935 [10], and now it is widespread in Europe (Greece, Russia), Asia (China, India, Israel, Japan, Korea, Pakistan), Canada, the USA, and Australia [8,9,11,12]. The resistance genes to CGMMV in cucumber are poorly investigated, and there are no commercially available resistant varieties. The only measure to protect plants against this virus is inoculation of young healthy plants with attenuated CGMMV strains that infect the host but do not induce visible symptoms, providing resistance to severe pathogen strains [9]. Therefore, it is of a particular interest to study the defense response mechanisms of cucumber plants to CGMMV infection.

Currently, one of the main approaches to investigating the systemic response of a particular organism to some external stress is transcriptome profiling. In recent years, the transcriptomes of more than 1000 plant species have been sequenced [13], and the data obtained were already used, for example, to analyze the virus-induced gene-silencing system [14], the repertoire of defense peptides [15], or the diversity of cytochrome P450 sequences [16] of various organisms. Paired sequencing of transcriptomes of healthy and infected plants gives useful data for the investigation of plant response to a specific pathogen and provides insight into the defense mechanisms involved in this response. For example, RNA-seq is used to study differential gene expression in plant response to viral infection [17]. A total of 1621 differently expressed genes (DEGs) were identified in CGMMV-inoculated watermelon at late stage of infection, among which 1052 were up-regulated [18]. Several DEGs were involved in carbohydrate metabolism, hormone biosynthesis and signaling transduction, secondary metabolites biosynthesis, and plant–pathogen interactions [18]. In order to better understand the molecular mechanisms of *Hop stunt viroid* (HSVd) infection in *Prunus avium* L. fruit, transcriptome analysis of HSVd-infected and healthy fruits was carried out. A total of 1572 DEGs were identified; of them, 961 were up-regulated. Functional analysis indicated that the DEGs were mainly involved in plant hormone signal transduction, plant–pathogen interactions, secondary metabolism, and the MAPK signaling pathway [19].

In the present study, the inbred line 229 of *Cucumis sativus* L. (Federal Scientific Vegetable Center (FSCV), Russia) highly susceptible to CGMMV infection was chosen for transcriptome analysis. Using next-generation sequencing (NGS) technology, we analyzed transcriptomes of cucumber plants inoculated with a pathogenic CGMMV strain at two time points (early infection 3 days post inoculation (dpi), later infection at 20 dpi) in comparison with those obtained from healthy plants at the same time points. We conducted in silico differential gene expression analysis for the pairs of healthy-infected transcriptomes. We focused on the analysis of SA-, JA-, and ET-dependent responses and investigated the transcriptional changes in PR-protein and TF genes activated by biotic stress. We also explored DEGs playing a crucial role in VIGS. As a result, we revealed the major molecular responses to CGMMV infection in cucumber.

## 2. Materials and Methods

### 2.1. Plant Samples, CGMMV Treatment and Confirmation of Viral Infection

A highly inbred line 229 of *C. sativus* L. (FSCV, Moscow region, Russia) susceptible to CGMMV infection was used for transcriptome analysis. Cucumber plants were grown in a laboratory greenhouse under controlled temperature and light conditions (25 °C day/18 °C night; 16-hour day/8-hour night). Two-week-old seedlings at the two fully expanded leaf stage were taken for experiments. The plants were divided into two groups: the first one included control healthy plants, while the plants of the second group were inoculated with 0.5 mg/mL of the pathogenic MC-1 strain of CGMMV [20]. Samples of leaves of the upper tier, located above the inoculated ones and infected due to systemic spread of the virus at 3 dpi and 20 dpi (no. 3i and 20i), as well as those from the healthy plants (no. 3h and 20h) at the same time points were taken for RNA isolation. Each 100 mg sample for transcriptome analysis was obtained from leaves of five different plants. The presence of CGMMV in leaf tissues was confirmed by RT-PCR using virus-specific primers (5473-dir and 5650-rev; Appendix A). For positive control, the housekeeping gene specific primers of protein phosphatase 2A regulatory subunit A (PP2A; NCBI Ref Seq: Csa_5G608520/XM_011657465.2) were used (PP2A-dir and PP2A-rev; Appendix A).

### 2.2. RNA Isolation

One milliliter of Intact RNA fixative (Evrogen, Moscow, Russia) was added to each sample for maintaining RNA integrity, and then the samples were stored at +4 °C. RNA was isolated in two technical replicates for each sample using ExtractRNA kit (Evrogen, Moscow Russia) followed by MiniElute PCR Purification kit (Qiagen GmbH, Hilden, Germany). The quality of the RNA samples obtained was evaluated using NanoDrop2000 (Thermo Fisher, Waltham, MA, USA) and Agilent 2100 Bioanalyzer (Agilent, Santa Clara, CA, USA). Four total RNA samples were used for preparation of cDNA libraries for subsequent Illumina sequencing.

### 2.3. Library Preparation and Sequencing

cDNA libraries for NGS were prepared according to the manufacturer’s instructions (Illumina, San Diego, CA, USA). Four mRNA samples were purified using RNA purification beads, followed by fragmentation and priming for cDNA synthesis. The SuperScript Double-Stranded cDNA Synthesis kit (Invitrogen, Waltham, MA, USA) was used for double-stranded cDNA synthesis, followed by purification on Agencourt AMPure XP beads (Beckman Coulter, Inc., Brea, CA, USA). Upon end repairing and 3′-ends adenylation, the RNA adapters’ ligation and enrichment of DNA fragments were performed. The library templates obtained were validated using an Agilent 2100 Bioanalyzer. Clonal clusters from DNA library templates were produced using a TruSeq PE Cluster Kit v2 and cBot automated system (Illumina, USA). Clusters obtained were finally used to perform paired-end sequencing on the Illumina Hiseq 2500 platform (San Diego, CA, USA), and 150 bp paired-end reads were obtained. Illumina sequencing was performed using the equipment of Evrogen, Russia.

### 2.4. Transcript Abundance Estimation and Differential Expression Analysis

Paired-end reads were subjected to Q15 filtering, removal of library adapter sequences, A/T stretches, and short reads (less than 50 bp). In order to perform the differential expression analysis, the filtered reads were mapped to the cucumber genome from Ensembl, release 41 (ftp://ftp.ensemblgenomes.org/pub/plants/release-41/fasta/cucumis_sativus, ASM407v2, GCA_000004075) using Bowtie2, and the counts for the annotated loci from this genome were calculated using HTSeq-count. Finally, the differential expression analysis was conducted using the DESeq2 R package [21] and several custom scripts for processing and presenting the data in concise and human-readable form [22,23]. The differentially expressed mRNAs and genes were determined with the following cutoffs: log2 (fold change) ≥ 1 or log2 (fold change) ≤ −1 and *p*-value < 0.05. Raw sequence reads for all samples were deposited to NCBI SRA archive (https://www.ncbi.nlm.nih.gov/sra) under project number PRJNA646644, SRS7015510–SRS7015513. The samples in SRA archive were named as 2hl, 5pt, 9hl, and 11pt. Here the samples were re-numbered for clarification purposes as 3h, 3i, 20h, 20i, respectively, where 3 and 20 denote the number of days post infection, ‘h’ stands for ‘healthy’ and ‘i’ stands for ‘infected’.

### 2.5. Quantitative RT-PCR (qRT-PCR)

Total RNA from different samples of cucumber leaves were isolated as described above. cDNA was obtained from total RNA by Thermo Scientific Revert Aid First Strand cDNA Synthesis Kit (Thermo Fisher Scientific, Waltham, MA, USA), and qRT-PCR was performed using qPCRmix-HS SYBR (Evrogen, Moscow, Russia), following the manufacturer’s protocols. All gene expression analyses were run in triplicate. The primers used for qRT-PCR are presented in Appendix A. The relative levels of transcripts were normalized to the expression level of the PP2A housekeeping gene, which was calculated in terms of threshold cycles using the 2^-∆∆Ct^ method.

## 3. Results

### 3.1. CGMMV Infection Confirmation and Plant Phenotypes

The highly inbred line 229 of *C. sativus* (FSCV, Moscow region, Russia) susceptible to CGMMV infection was inoculated with the pathogenic MC-1 CGMMV strain. The inoculated plants showed symptoms of infection at 10 dpi. At 3 dpi, the infected plants showed no visible symptoms of the disease (Figure 1A); however, RT-PCR analysis clearly demonstrated the presence of the virus in inoculated leaves (Figure 1B). At 20 dpi, the infected plants showed obvious disease symptoms, in particular, specific green mosaic on leaves (Figure 1A), and RT-PCR with CGMMV-specific primers confirmed the presence of the virus (Figure 1B).

### 3.2. Overview of the RNA-seq Results

The first step of sequencing data analysis included collecting general statistics to make quality estimations and data suitability for the studies planned. General statistics of transcriptome sequencing on the Illumina platform are given in Table 1. The percentage of reads mapped back to the assembly was equal to 96−98% for all samples. Together with high median length and good coverage (more than 40×), this ensures the high quality of transcriptome sequencing that allows performing reliable annotation and differential expression estimations.

We also performed the functional annotation of transcriptomes using GO (gene ontology) terms. The chart showing the fractions of annotated transcripts assigned to various biological processes is presented in the Figure 2. This distribution was essentially the same for all four transcriptomes. Since the aim of this work was to study genes involved in the defense response of cucumber plants to the pathogen, the transcripts attributed to ‘Immune system process’ and ‘Response to stimulus’ received primary attention in further DEGs studies.

### 3.3. DEGs in Response to CGMMV-Induced Stress

Genes with a false discovery rate (FDR) < 0.05 and an estimated absolute log2 fold change (log2FC) ≥ 1 in sequence counts between libraries were considered significantly differentially expressed. Only the genes having normalized counts ≥ 10 for both conditions, as estimated by DESeq2, were analyzed further in order to increase the reliability of the results.

For the first pair of healthy and infected plant samples (3 dpi, no. 3h and 3i, respectively), 5766 DEGs were identified according to these criteria (Appendix A), of which 2536 were down-regulated and 3230 were up-regulated after CGMMV infection. For the second pair of samples obtained at 20 dpi (no. 20h and 20i, respectively), the numbers of down- and up-regulated DEGs upon infection were 2022 and 2366, respectively, giving 4388 DEGs in total (Appendix A). The volcano plots of DEGs for both pairs of samples are shown in Figure 3. Volcano plots show that the distributions of up- and down-regulated genes were similar for both pairs of samples; however, in the second pair, this distribution was slightly shifted towards the set of down-regulated genes.

Another useful tool for making comparison between several sets of genes are Venn diagrams, which are presented in Figure 4. They reveal the complex nature of expression changes caused by CGMMV infection. A total of 3514 DEGs were discovered only at 3 dpi, while 2138 were revealed only at 20 dpi, and 2238 were found in both cases. Of these 2238 common genes, 241 were up-regulated in infected plants both at 3 dpi and 20 dpi, 55 genes were down-regulated at both time points, and each of the remaining 1942 genes were up-regulated at one time point but down-regulated at another one. Thus, only about 30% of DEGs were revealed at both time points, and most of them changed their expression pattern at different stages of the infection process. We used the functional annotation and signaling pathway information to elucidate the possible defense mechanisms reflected by these complex patterns of expression changes.

The functional groups of transcripts mapped to 10 top genes that showed the maximum changes in transcription levels induced by CGMMV infection were also analyzed. These gene subsets are presented in Table 2. A complete list of DEGs including the counts, fold change, and FDR is given in Appendix A. Subsets of DEGs belonging to various pathways and families described here are presented in Appendix A.

At the early stage of infection, four annotated DEGs were induced dramatically (log2(FoldChange) ≥ 7.2)—namely, DEGs encoding histones 2, 3, 4, and benzylalcohol O-benzoyl transferase (BEBT)—while 3 DEGs encoding metal-tolerance protein (MTP), proline-rich protein 1(PRP1), and sucrose synthase (SuSy) were strongly down-regulated. The maximum repression was exhibited by the DEG Csa_3G105950 encoding MTP, which is a divalent cation transporter essential in general metal homeostasis and tolerance to metal excess [24]. Four strongly induced DEGs histones known to play a crucial role in gene expression regulation. DEG encoding BEBT (Csa_2G429030) was dramatically up-regulated at 3 dpi and down-regulated at 20 dpi (log2(FoldChange) = −6.7), as well as the DEG Csa_4G308490 encoding BAHD acyltransferase (log2(FoldChange) = −6.4). BEBT and BAHD acyltransferase are involved in biosynthesis of benzenoids, benzoic acid, and phenolic compounds [25].

Drastically down-regulated DEG Csa_2G176690 encodes a proline-rich protein, which is a structural cell wall protein involved in a number of developmental processes from germination to plant death [26]. The DEG Csa_4G001950 was significantly down-regulated (log2(FoldChange) = −7.1); it encoded sucrose synthase (SuSy), a key enzyme in sucrose metabolism, primarily in sink tissues. SuSy plays a crucial role in such metabolic pathways as energy production, primary-metabolite production, and the synthesis of complex carbohydrates. Plants with reduced SuSy activity exhibited stunted growth, modified leaf morphology, and shoot apical meristem function [27].

At 20 dpi, most of 10 top DEGs were dramatically repressed, except for Csa_2G176190 encoding a repetitive PRP3 and Csa_6G448740 encoding a chloroplastic threonine dehydratase (TD). TD catalyzes the conversion of threonine into α-ketobutyrate and ammonia in isoleucine biosynthesis [28]. The DEG Csa_4G154320 encoded a polygalacturonase-inhibiting protein (PGIP)—a conserved cell wall protein found in monocot and dicot plants, which inhibits the activity of polygalacturonases secreted by microorganisms and pests [29]. This DEG was dramatically down-regulated at 20 dpi (log2(FoldChange) = −7.3), which may indicate that this type of defense response was suppressed in CGMMV-infected cucumber plants. The DEG Csa_1G050360 encoded glyoxysomal-like malate synthase (MLS) involved in the glyoxylate cycle: MLS catalyzes the synthesis of malate from glyoxylate and acetyl-CoA. MLS activity was reported to increase in *Arabidopsis* seedlings after imbibition and decrease after germination, and expression of the MLS gene was induced during sugar starvation [30]. This DEG was dramatically repressed at 20 dpi (log2(FoldChange) = −6.3) in cucumber plants infected with CGMMV. The DEG Csa_3G076580 was strongly repressed as well; it encoded a plastid lipid-associated protein that is accumulated in fibrillar-type chromoplasts. This protein was reported to be associated with the protection of thylakoid membranes upon biotic stress, in particular, bacterial infections [31].

Concluding, at 3 dpi, five of the eight annotated Top10 DEGs displayed a remarkable increase in expression level—in particular, the genes encoding histones, BEBT, and BAHD acyltransferase. This could indicate a defense response of cucumber plants to CGMMV infection because histones were shown to play an important role in expression regulation of various genes, while BEBT and BAHD acyltransferase were demonstrated to induce synthesis of secondary metabolites and other defense compounds [32]. At the same time, the plant–pathogen interaction studied was compatible, and the viral invasion caused severe suppression of the genes involved in stress response and plant development, including DEGs encoding MTP, PRP, and SuSy. At 20 dpi, the cucumber plants showed visible symptoms of CGMMV infection and, at this stage, almost all Top10 DEGs displayed drastic repression, except the DEG encoding the cell wall PRP3. DEGs encoding TD, PGIP, BEBT, BAHD acyltransferase, MLS, and plastid lipid-associated protein were significantly down-regulated, which resulted in reduced resistance to biotic stress, changes in leaf morphology, modifications of thylakoid membranes, and repression of secondary metabolite biosynthesis.

### 3.4. DEGs Involved in Host Defense Response

In this study, we analyzed SA-, JA-, and ET-dependent pathways associated with defense response against CGMMV and changes in expressions of PR-proteins, TFs, and proteins involved in VIGS. The list of transcripts involved in signaling pathways was selected according to the KEGG PATHWAY Database (https://www.kegg.jp/kegg/pathway.html, accessed on 1 March 2021). In total, we revealed 108 DEGs involved in plant defense response. Among them, 34 DEGs belonged to SA biosynthesis and signaling, including 10 DEGs involved in the phenylalanine ammonia-lyase (PAL) pathway, 1 DEG encoding an enhanced disease susceptibility 1 (EDS1) protein, 1 DEG encoding a phytoalexin deficiency 4 protein (PAD4), 14 DEGs encoding non-race-specific disease resistance 1 proteins (NDR1), 2 DEGs for non-expressor of pathogen-related genes 1 (NPR1), 5 DEGs for various TFs, and 1 DEG encoding chalcone synthase (CHS). A total of 11 DEGs playing an essential role in ethylene response were identified, of which 3 DEGs encoding aminocyclopropane-1-carboxylic acid oxidase (ACO), 1 DEG encoding CONSTITUTIVE RESPONSE1 (CTR1) kinase, 2 DEGs encoding EIN3-BINDING F-BOX1 and 2 (EBF1/2) proteins, 3 DEGs encoding ethylene-response transcriptional factors (ERF), 1 DEG encoding mitogen-activated protein kinase 9 (MKK9), and 1 DEG encoding MKK5. In addition, 7 DEGs associated with JA response were revealed—namely, 5 DEGs encoding jasmonate-ZIM domain repressor proteins (JAZ), 1 DEG encoding jasmonate amino acid synthetase 1 (JAR), and 1 DEG encoding MYC2 TF. We also identified 9 DEGs related to VIGS, including 5 DEGs encoding RNA-dependent RNA polymerases (RDRP), 1 DEG encoding Dicer-like (DCL) enzyme, and 3 DEGs encoding Argonaute (AGO) proteins. We found 9 DEGs encoding different PR-proteins, among them 1 DEG encoding a PR-1 protein, 3 DEGs encoding PR-2 proteins, 1 DEG encoding a PR-3 protein, 1 DEG encoding a PR-4 protein, and 3 DEGs encoding PR-8 proteins. Finally, we analyzed 46 differentially expressed genes encoding TFs, which belonged to 7 different families—in particular, 17 DEGs encoding WRKY TFs, 13 encoding NAC TFs, 6 encoding MYB TFs, 3 encoding bZIP (TGA, HBP-1b) TFs, 3 encoding ERF TFs, 2 encoding bHLH (MYC) TFs, and 2 encoding TCP TFs (see below).

### 3.5. RNA-seq Validation by qRT-PCR

Nine genes were selected for verification of changes in gene expression levels revealed by RNA-seq based on the following considerations. The selected genes belonged to different functional groups: genes encoding proteins involved in signaling pathways—CHS (Csa_3G600020), ACO1 (Csa_6G421630), PAL (Csa_6G445760), NPR1 (Csa_4G063470); TF genes induced by biotic stress—MYC3 (Csa_3G002860), WRKY (Csa_3G730800), ERF C3 (Csa_3G135120) and ERF069 (Csa_6G133770); and genes involved in RNA-silencing—DEG encoding RDRP (Csa_1G005580); their minimum expression level according to transcriptome analysis data should be at least 50 counts. qRT-PCR analysis was performed for all of these genes, and the relative expression levels for them are shown in the Appendix A. qRT-PCR data have confirmed our RNA-seq results.

## 4. Discussion

### 4.1. Salicylic Acid Biosynthesis and Signaling

Salicylic acid is a key hormone in plant defense signaling. It is responsible for the development of local response and systemic acquired resistance to a wide range of pathogens including bacteria, fungi, and oomycetes [33]. In plants, SA is synthesized from chorismate via two different pathways: the isochorismate and the phenylalanine ammonia-lyase (PAL) pathways (Figure 5). The first pathway works in chloroplasts: isochorismate synthase 1 (ICS1) catalyzes conversion of chorismate to isochorismate, and then isochorismate is converted to SA by isochorismate pyruvate lyase [33]. In *Arabidopsis thaliana,* the following proteins play an important role in the ICS1 pathway: NDR1, EDS1, PAD4, and a family of transcription factors (TFs), such as TCP (TEOSINTE BRANCHED 1, CYCLOIDEA, PCF1) TCP8 and TCP9 (Qi G_2018). EDS1 and PAD4 are essential components of effector-triggered immunity (ETI); they induce SA biosynthesis through up-regulation of ICS1 and can themselves be induced by SA. In cucumber transcriptomes, we found that one EDS1 DEG (Csa_1G006320) was down-regulated at 3 and 20 dpi in infected plants, and one PAD4 DEG (Csa_4G496760) was down-regulated at 3 dpi (Appendix A). NDR1 is a GPI-anchored membrane protein that is involved in the regulation of SA accumulation and is one of the important players in ETI since it interacts with R-proteins. In cucumber transcriptomes, 14 NDR1 DEGs were found. At 3 dpi, six of them (Csa_5G139020, Csa_1G064760, Csa_3G418790, Csa_6G042460, Csa_7G451320, Csa_3G779770) were significantly up-regulated in infected plants, and one DEG (Csa_4G269210) was slightly down-regulated. At 20 dpi, 11 NDR1 genes changed expression, 4 (Csa_3G418790, Csa_3G779770, Csa_3G209460, Csa_3G116630) were down-regulated, and 7 (Csa_4G269210, Csa_6G042460, Csa_1G169420, Csa_6G425840, Csa_3G779010, Csa_3G779020, Csa_1G601000) were up-regulated, including Csa_6G425840, whose expression increased dramatically (log2(FoldChange) = 4.8) in comparison with non-infected plants (Figure 5, Appendix A). We also revealed two DEGs of TFs, TCP8 and TCP9, which were known to be the positive regulators of SA biosynthesis because they facilitated ICS1 expression during plant defense [33]. TCP9 (Csa_6G075180) was significantly up-regulated (log2(FoldChange) = 2.6) at 3 dpi, and by 20 dpi, it was strongly down-regulated (log2(FoldChange) = −2.4), while TCP8 (Csa_5G587110) did not change its expression level in infected plants at the early stage of invasion and was slightly up-regulated at the late stage (Figure 5, Appendix A).

The second way of SA biosynthesis is the PAL pathway occurring in the cytoplasm. In short, it begins with the formation of prephenic acid from chorismate, then reduced to phenylalanine and consequently to cinnamate by PAL. The cinnamate is converted to ortho-courmaric acid or benzoate, both of which are SA precursors [34] (Figure 5, only few steps are shown). In addition to SA formation, PAL is involved in phenylpropanoid biosynthesis and the biosynthesis of secondary metabolites [34]. In the present study, we found 10 differently expressed genes encoding PAL. At the early stage of infection, six genes (Csa_4G008760, Csa_4G008770, Csa_6G445240, Csa_6G445750, Csa_6G445760, Csa_6G445770) were significantly down-regulated, while three cucumber PAL transcripts (Csa_6G147460, Csa_6G405960 and Csa_6G445740) were up-regulated (Figure 5, Appendix A). At the late stage, all detected PAL genes were up-regulated (Appendix A).

Once SA is synthesized, it activates NPR1, a master regulator of SA-mediated plant defense [33]. NPR1 was shown to control expression of the majority of SA-regulated genes. Pathogen invasion or SA treatment induces conversion of NPR1 in the cytoplasm from oligomers to monomers, which move to the nucleus, interact with TFs, and induce expression of genes encoding PR-proteins with antimicrobial activity that leads to disease resistance. Two NPR1 DEGs were found in *C. sativus* in response to CGMMV in our experiments. Csa_4G063470 was detected only at 3 dpi, and it was slightly down-regulated in infected plants; Csa_3G822220 was found at 20 dpi and it was up-regulated (Figure 5, Appendix A).

Another enzyme that plays an important role in plant immunity in response to abiotic stress and pathogen invasion is chalcone synthase. This is a key protein of the flavonoid biosynthetic pathway and biosynthesis of secondary metabolites; it is responsible for accumulation of flavonoids, lignins, and phytoalexins and is involved in the salicylic acid pathway [35]. We revealed one DEG (Csa_3G600020) encoding CHS, and its expression level was dramatically increased (log2(FoldChange) = 5.1) at 3 dpi and strongly decreased at 20 dpi (log2(FoldChange) = −1.9) in plants inoculated with CGMMV (Appendix A).

Among the major TF players in SA response, we found three DEGs encoding bZIP TFs. This type of TF contains a bZIP domain with a basic region for DNA binding and a leucine zipper region. In *Arabidopsis*, bZIP proteins include TGA TFs representing key regulators of SA signaling [36]. In maize and tobacco, expression of bZIP TFs increased upon inoculation with the fungus *Ustilago maydis* and Potato virus X [4]. In our study, we observed that three DEGs encoding TFs—namely, TGA-1 (Csa_4G036580), TGA-7 (Csa_2G403160) and HBP-1b (Csa_3G819960)—were down-regulated at the early stage of infection (Figure 5, Appendix A), and one of them, TGA-1, was slightly up-regulated at the late stage. These results suggest that bZIP TFs, except for TGA-1, are not involved in the immune response of cucumber plants to the CGMMV infection.

Concluding, at the early stage of infection, the majority of NDR1 DEGs playing a significant role in the ICS1 pathway were up-regulated, while most PAL DEGs were down-regulated. In other words, the ICS1 pathway was activated, while the PAL pathway was suppressed in response to CGMMV infection. At the late stage, most NDR1 DEGs were up-regulated, but some were down-regulated, while all PAL DEGs were strongly up-regulated, suggesting that at 20 dpi, the PAL pathway played a role in SA biosynthesis and SA-mediated plant defense.

### 4.2. Ethylene Biosynthesis and Signaling

Ethylene is a gaseous phytohormone that regulates various aspects of plant development and plays an essential role in adaptive responses to biotic and abiotic stresses [37]. Ethylene biosynthesis starts with methionine, which is first converted to S-adenosyl-L-methionine (SAM) by SAM synthase. SAM is then converted to 1-aminocyclopropane-1-carboxylic acid (ACC) by the enzyme ACC synthase (ACS), and finally ethylene is generated from ACC by ACC oxidase (ACO) in the presence of oxygen [37]. ACO is encoded by a number of genes that are differentially expressed in response to pathogen attack in particular, and up-regulation of ACO can activate ethylene production and lead to enhanced resistance [38]. In cucumber transcriptomes, we found three ACO DEGs; one of them (Csa_4G361270) with a low expression level was slightly suppressed by CGMMV infection at 3 dpi. Two others (Csa_6G160180 and Csa_6G421630) were strongly expressed. The Csa_6G421630 gene was up-regulated at the early stage of infection and then repressed, while Csa_6G160180, conversely, was first down-regulated and then induced (Figure 6, Appendix A).

Ethylene signaling begins with perceiving the molecule by several specific receptors located in the ER membrane. We found one DEG (Csa_3G141850) encoding the ETR2 receptor that was down-regulated at 20 dpi in infected plants (Appendix A). Ethylene binding disrupts the interaction between the receptor and a serine/threonine protein kinase named CTR1 that is a negative regulator of ethylene response [39]. In our experiments, one DEG (Csa_6G450400) encoding CTR1 was down-regulated at 3 dpi in infected plants; this repression might lead to induction of the ethylene pathway during the early stage of infection. CTR1 phosphorylates a substrate ETHYLENE-INSENSITIVE2 (EIN2) that interacts with ethylene receptors. When receptors bind ethylene, CTR1 becomes inactivated and the C-terminal fragment of EIN2 is released and translocated to the nucleus, where it suppresses the translation of two proteins EBF1 and 2 [39]. Two DEGs (Csa_2G006790, Csa_3G878200) encoding EBF1/2 with constitutively high expression level were revealed in the cucumber plants in response to CGMMV infection (Appendix A). Both of them were strongly up-regulated at 3 dpi and dramatically down-regulated at 20 dpi. EBF1/2 controls the proteolytic degradation of EIN3 and EIL1, two key transcription factors of the ethylene signaling pathway; when EBF1/2 is suppressed, TFs accumulate in the nucleus and activate the transcription of the ethylene response genes, including the ethylene-response factor (ERF) [39]. In cucumber plants inoculated with CGMMV, three ERF DEGs (Csa_2G177210, Csa_3G135120, and Csa_6G133770) with high expression level were found. Two of them were significantly up-regulated at 3 dpi and one (Csa_6G133770) was down-regulated. At 20 dpi, Csa_2G177210 and Csa_3G135120 were strongly down-regulated, while Csa_6G133770 was up-regulated (Figure 6, Appendix A).

In addition, mitogen-activated protein kinases (MAPKs), the enzymes catalyzing phosphorylation of protein substrates by serine or threonine residues, are also involved in ET biosynthesis and signaling by controlling transcription of ACS [40]. The MKK9 (mitogen-activated protein kinase kinase 9) is involved in the regulation of ET biosynthesis and signaling, acting downstream of CTR1. We found one DEG (Csa_1G042980) encoding MKK9 with high level of expression that was significantly up-regulated in infected plants at 3 dpi. Moreover, we revealed one DEG (Csa_3G651720) encoding the kinase MKK5 that acts upstream of MPK3/MPK6. It had a low expression level and was slightly up-regulated in infected plants at 20 dpi (Figure 6, Appendix A).

Concluding, ethylene signaling was strongly induced in response to CGMMV infection at the early stage of infection. At 3 dpi, six DEGs with high initial expression level were significantly up-regulated, including a 10- and almost 20-fold increase in EBF1/2 DEGs. At the late infection stage, ethylene response was drastically repressed: seven out of nine highly expressed DEGs were significantly down-regulated, especially EBF1/2 DEGs, whose expression was reduced by more than 20 times. Interestingly, ACO DEG (Csa_6G160180) was, in contrast, strongly down-regulated at 3 dpi and up-regulated at 20 dpi, which evidently influenced ethylene production.

The expression of genes encoding ACO, key enzymes of ethylene biosynthesis, were also studied in the work devoted to the transcriptome analysis of watermelon response to CGMMV infection [18]. Four genes of ACC oxidases were differentially expressed in response to CGMMV infection. At the late development stages of watermelon fruits, two DEGs encoding ACO were induced; the rest were repressed. The authors concluded an induction of ET biosynthesis and signaling pathways, which complements our data on the development of the antiviral response in the earlier stages of CGMMV infection.

### 4.3. Jasmonic acid Biosynthesis and Signaling

Jasmonic acid is a one of three classical defense phytohormones, which, in addition to SA and ET, play an important role in regulating plant defense responses to pathogens. JA biosynthesis develops in three stages, which take place in the plastid, peroxisome, and cytosol (Figure 7) [41].

Briefly, pathogen attack induces activation of phospholipase in the plastid membrane and promotes synthesis of linolenic acid, it being a JA precursor. Linolenic acid is converted to 12-oxo-phytodienoic acid (OPDA) through several steps, including oxygenation with various enzymes [41]. In peroxisome, OPDA is converted to JA by OPDA reductase through reduction and oxidation steps. In cytoplasm, JA is converted to a number of derivatives, including the major bioactive form of JA: cis-jasmone, methyl jasmonate (MeJA), and jasmonyl isoleucine. Jasmonyl isoleucine is reversibly converted from JA by jasmonate amino acid synthetase 1 (JAR1). In cucumber, we found that JAR1 DEG (Csa_3G119760) was present only at 20 dpi, and its moderate expression level was suppressed in cucumber plants infected with CGMMV (Figure 7, Appendix A). Jasmonyl isoleucine is transported to the nucleus where it is recognized and perceived by coronatine insensitive 1 protein (COI1), providing the binding to jasmonate-ZIM domain repressor protein (JAZ). COI1, being a part of the F-box ubiquitin complex, initializes the destruction of JAZ and the release of multiple transcription factors of JA response genes that were suppressed by JAZ [41]. In cucumber transcriptomes, five DEGs encoding JAZ were revealed. Two of them (Csa_1G042920, Csa_6G091930) were up-regulated at 3 dpi and down-regulated at 20 dpi in infected plants. One DEG (Csa_7G448810) was found only at the early stage, where it was up-regulated by infection. At the late stage of infection, two other DEGs were identified; one (Csa_1G597690) with high expression level was suppressed, and another (Csa_3G645940) was up-regulated. JAZ repressor interacts directly with MYC2 TF, which induces a set of JA-responsive genes [42]. At 20 dpi, we found another DEG (Csa_3G902270) encoding MYC2 that had a moderate expression level and was significantly suppressed in infected plants (Figure 7, Appendix A).

Concluding, CGMMV infection did not evoke a JA-dependent defense response in cucumber plants because only seven DEGs playing a role in JA signaling were found in cucumber transcriptomes. At 3 dpi, just three DEGs encoding JAZ proteins were up-regulated, while at 20 dpi, all but one DEG were strongly down-regulated.

### 4.4. Expression of PR-Protein Genes in Response to CGMMV Infection

PR-proteins are structurally diverse polypeptides that are considered to be important components of plant innate immunity. These proteins are divided into 17 distinct families sharing one common feature: their genes are induced in response to pathogen attack. Signaling molecules, such as SA, ethylene, and JA, also trigger accumulation of these proteins. The SA-dependent pathway induces transcription of PR-1, PR-2, and PR-5 protein genes, leading to accumulation of the proteins locally or systemically providing the systemic acquired resistance. JA induces accumulation of PR-3, PR-4, and PR-12 proteins locally and facilitates the development of local acquired resistance [3].

PR-1 proteins are widely used as molecular markers of systemic acquired resistance response—in particular, of the SA-dependent pathway. Expression of PR-1 protein genes is induced in response to a variety of pathogens [43]. We found no PR-1 DEGs in cucumber transcriptomes, but one gene (Csa_1G420360) encoding a homologue of PR-1 protein, which did not belong to known signaling pathways described in KEGG, was repressed at 3 dpi in infected cucumber plants (Appendix A).

PR-2 proteins belong to β-1,3-glucanases, the enzymes that cleave the polysaccharides of bacterial and fungal cell walls [44]. In the *C. sativum* genome, PR-2 proteins are encoded by a multigene family. In cucumber plants, three genes changed expression in response to CGMMV infection. Two DEGs encoding PR-2 proteins were weakly expressed; one of them (Csa_1G660200) was slightly up-regulated at 3 dpi in infected plants, and another (Csa_1G616240) was down-regulated at early and late stages of infection. The third gene (Csa_5G181480) was strongly expressed and was up-regulated in CGMMV-inoculated plants at 3 dpi (Appendix A).

PR-3 proteins are extracellular chitinases that hydrolyze chitin of the fungal cell walls [45]. Only one DEG (Csa_6G508020) encoding a chitinase was found in this study. It had a very low expression level and was slightly repressed in CGMMV-infected plants at 3 dpi. We also found one homologous protein, a chitinase-like protein 1 (Csa_4G017110), which was strongly induced in infected plants at the early stage of infection (Appendix A).

PR-4 proteins belong to endochitinases with a conserved C-terminal BARWIN domain. PR-4 proteins are divided into class I proteins sharing a conserved N-terminal chitin-binding hevein-like domain and class II proteins that lack this domain [46]. We revealed one DEG (Csa_1G534750) considered to be a class I PR-4 protein. It had a moderate expression level, which was induced at 3 dpi and strongly up-regulated at 20 dpi in infected cucumber plants (Appendix A).

PR-8 proteins represent class III chitinases induced by SA and pathogen invasion [47]. Expression of PR-8 genes in cucumber plants increased significantly after systemic acquired resistance inducer exposure and in response to biotic stress [48]. We found three DEGs encoding PR-8 proteins, two of which (Csa_6G338110 and Csa_4G082450) had a low expression level, while one DEG (Csa_7G318990) was strongly expressed. Csa_7G318990 was strongly suppressed at 3 dpi in infected plants and slightly up-regulated at 20 dpi. Csa_6G338110 was up-regulated at 3 dpi and down-regulated at 20 dpi, while Csa_4G082450, conversely, was down-regulated at the early stage and slightly up-regulated at the late stage (Appendix A).

Concluding, genes encoding PR-proteins insignificantly changed their expression in response to CGMMV infection. Among numerous PR-protein genes, only 10 DEGs were revealed, and all of them responded to infection at 3 dpi; half of them were down-regulated, and half were up-regulated. At 20 dpi, only five DEGs were revealed, four of which were up-regulated. It should be mentioned that PR-protein DEGs were mostly poorly expressed; only three DEGs encoding PR-3 (Csa_4G017110), PR-4 (Csa_1G534750), and PR-8 (Csa_7G318990) changed their expression level significantly. Thus, defense response to CGMMV infection was not associated with massive up-regulation of PR-protein genes. 

### 4.5. Transcription Factors Associated with Biotic Stress

A number of TFs were reported to play a key role in reprogramming transcription of host cells in response to pathogen attack. A total of 1931 TF sequences (1403 loci) belonging to 57 TF families were revealed in cucumber plants according to the PlantTFDB v5.0 (http://planttfdb.gao-lab.org/index.php?sp=Csg, accessed on 1 March 2021) data base. In this study, we focused on TFs involved in regulation of plant–pathogen interactions and plant hormone signal transduction. As a result, 46 differentially expressed genes of TFs belonging to seven different families were revealed in cucumber plants in response to CGMMV infection at 3 and 20 dpi. A total of 17 DEGs encoded WRKY TFs, 13 encoded NAC TFs, 6 encoded MYB TFs, 3 encoded bZIP (TGA, HBP-1b) TFs, 3 encoded ERF TFs, 2 encoded bHLH (MYC) TFs, and 2 encoded TCP TFs (Figure 8). In comparison, Sun et al. identified 59 DEGs encoding TFs involved in the corresponding pathways in watermelon leaves at 24 h post-inoculation with CGMMV [49]. Note that the TFs bZIP (TGA, HBP-1b), TCP, and ERF played a key role in the SA and ethylene pathways, respectively, and were discussed in previous sections.

WRKY TFs belong to one of the largest TF families and are regarded as the major regulators involved in MAMPs- and effector-triggered immunity. These proteins comprise a zinc finger motif at the C-terminus and a DNA-binding domain containing a conserved WRKY-domain at the N-terminus. WRKYs were found to play an important role in plant–pathogen interactions for various plants [50]. CaWRKY1, a WKRY protein from chili pepper, negatively regulates the defense mechanism because virus-induced gene silencing of this factor leads to inhibition of growth of *Xanthomonas axonopodis,* while its overexpression provokes an enhanced hypersensitive response to *Pseudomonas syringae* and Tobacco mosaic virus [51]. Genes of six WRKY TFs found in tomato were responsive to Tomato yellow leaf curl virus infection [52]. WRKY factors of *Medicago truncatula* were reported to be involved in elicitor-triggered reprogramming of secondary metabolite biosynthesis, lignin deposition, PR gene expression, and systemic defense responses against Tobacco mosaic virus [53]. In this study, as many as 17 DEGs encoding WRKY TFs were found, and it was the largest family of TFs revealed in cucumber transcriptomes. In general, the expression of WKRY genes was inhibited at 3 dpi: only 2 genes were slightly up-regulated, while 10 genes were significantly down-regulated in response to CGMMV infection. In contrast, at 20 dpi, 13 genes, including those that showed no expression at the early stage were strongly up-regulated, and only 1 gene (Csa_3G121580) was slightly up-regulated (Figure 8, Appendix A).

Another family of plant biotic and abiotic stress-responsive transcriptional factors is NAC. These proteins consist of a conserved N-terminal DNA-binding NAC domain and a diverse C-terminal activation domain [54,55]. NAC TFs are involved in the regulation of various plant processes, including growth, development, and resistance to biotic and abiotic stresses [55]. Functional analyses of NAC TFs in knockout/knockdown mutants and overexpression lines of *Arabidopsis* and rice detected dozens of NAC genes involved in plant defense. They act as negative or positive regulators, modulators of hypersensitive response and stomatal immunity, or as targets of pathogen effectors [54]. For example, the transgenic rice line overexpressing *OsNAC6* showed enhanced resistance to *Magnaporthe grisea*-induced rice blast disease [56]. Cotton plants with silenced *GbNAC1* gene were susceptible to *Verticillium dahlia,* whereas *Arabidopsis* plants overexpressing the *GbNAC1* gene had increased resistance to this fungus [57]. In our study, 13 DEGs encoding NAC TFs in response to CGMMV infection were revealed, the majority of which were poorly expressed. At the early stage of infection, six genes were induced; two of them (Csa_4G361820 and Csa_6G382950) had a high expression level, and they were strongly up-regulated by the virus (Figure 8, Appendix A). Meanwhile at the same stage, four poorly and one highly expressed (Csa_6G127320) genes were down-regulated. At 20 dpi, only seven DEGs encoding NAC TFs were discovered—three of them were slightly up-regulated in response to CGMMV, while four others were down-regulated, including Csa_3G101810 with an extremely high level of expression and strongly expressed Csa_4G011770 (Figure 8, Appendix A).

MYC factors belong to a large bHLH superfamily that comprises the proteins with a conserved bHLH domain, including an N-terminal basic region and a C-terminal HLH region of 50 aa, which adopt a conformation with two alpha-helices separated by a loop [58]. bHLHs are involved in the regulation of such processes as seed germination, flowering time, abiotic stress responses, and biosynthesis of flavonoids through regulation of the phenylpropanoid pathway [53,58]. We found two DEG encoding MYC TFs playing an important role in JA signaling. DEG Csa_3G902270 encoding MYC2 and participating in JA-response was significantly down-regulated at 20 dpi, while gene of MYC3 TF (Csa_3G002860) was up-regulated at 3 dpi (Figure 8, Appendix A).

MYB TFs are functionally diverse proteins found in many eukaryotes, including plants. MYB TFs play an important role in biological processes, such as development, metabolism, and response to biotic and abiotic stresses. These proteins have different numbers of MYB DNA-binding domains consisting of approximately 50 amino acids that share a helix–turn–helix structure. The first identified MYB gene (‘*colored1*’) from *Zea mays* was involved in anthocyanin biosynthesis [59]. After this finding, a wide variety of MYB proteins involved in plant–pathogen interactions were discovered. For instance, in Arabidopsis, AtMYB30 was reported to play a role in regulation of the synthesis of very-long-chain fatty acids, that, in turn, could activate the hypersensitive cell death program [60]. The overexpression of rice *OsMYB4* induced the expression of a number of genes in *A. thaliana* and tomato involved in resistance to Tobacco necrosis virus, *Pseudomonas syringae*, and *Botrytis cinerea* [61]. In this study, we detected six DEGs encoding MYB TFs; all of them were expressed at 3 dpi and only two DEGs were found at 20 dpi. At the early stage, two DEGs with a low level of expression were up-regulated, and four DEGs with low or moderate expression level were down-regulated in cucumber plants inoculated with CGMMV. At the late stage, two poorly expressed MYB genes were slightly up-regulated (Figure 8, Appendix A).

In summary, among all the revealed genes of TFs, genes encoding TFs of WRKY and NAC families were the most sensitive to CGMMV infection. WRKY genes were significantly repressed at the early stage and induced at the late stages, while NAC genes were represented equally in both up- and down-regulated sets.

### 4.6. Silencing

One of the major plant responses to viral infection is RNA silencing. This system is triggered by double-stranded RNAs, the intrinsic moieties of the viral genome produced during virus replication by RNA-dependent RNA polymerases (RDRPs). The double-stranded RNAs are subsequently processed by Dicer-like enzymes into short 21–24 nucleotide RNA duplexes (small interfering RNAs, siRNAs), some of which can be species-specific [62]. The siRNAs, in turn, are loaded into AGO proteins and form an RNA-induced silencing complex (RISC). RISCs are targeted to viral RNAs that are complementary to siRNAs and induce post-transcriptional gene silencing or VIGS by endonucleolytic cleavage or translational repression [6]. In this study, we found five DEGs encoding cucumber RDRPs, three of which were down-regulated at 3 dpi in plants inoculated with CGMMV, including Csa_1G005580, which had a moderate expression level (Figure 9, Appendix A).

At 20 dpi, three detected DEGs of RDRP had a low expression level, two of which were slightly up-regulated and one that was insignificantly down-regulated. We also revealed three DEGs encoding different AGO proteins, one of which was slightly up-regulated at 3 dpi; however, its expression level was extremely low. Two other DEGs found at 20 dpi only were also weakly expressed and were slightly repressed in infected plants. Finally, we managed to find at 3 dpi one DEG (Csa_1G267180) encoding DCL enzyme that had a low expression level and was repressed in plants infected with CGMMV (Figure 9, Appendix A). Thus, the analysis of silencing DEGs showed that all 9 VIGS genes had low or moderate expression levels and were mainly repressed at the early and late stages of infection.

## 5. Conclusions

In this study, we analyzed transcriptomes of healthy and CGMMV-inoculated cucumber plants at the early and late stages of infection. In total, 5766 DEGs were found at 3 dpi, among which the most up-regulated by the biotic stress were the DEGs encoding histones and BEBT, while the most repressed were DEGs encoding MTP, PRP, and SuSy. At 20 dpi, 4388 DEGs were identified; the strongest repression was observed among DEGs encoding TD, PGIP, BEBT, BAHD acyltransferase, MLS, and a plastid lipid-associated protein, while the most inducible was the DEG encoding the cell wall PRP3. In our research, we mainly focused on SA, ET, and JA signaling and VIGS genes, as well as on genes encoding PR-proteins and TFs. The highest number of DEGs was revealed in the SA-biosynthesis pathway and signaling; furthermore, the majority of genes induced at 3 dpi encoded NDR1 proteins, while a high number of PAL pathway DEGs showed significant up-regulation at the late stage. We may speculate that at the early stage of infection, the ICS1 pathway was activated, while the PAL pathway played a pivotal role in SA biosynthesis at the late stage. In addition to SA-dependent defense signaling, the ethylene pathway was strongly induced by CGMMV at the early stage of infection and dramatically repressed at the late stage. In the ethylene pathway, we revealed fewer DEGs than in the SA-biosynthetic pathway and immune response; however, the level of expression of ET-associated DEGs and their changes were significantly higher. Surprisingly, VIGS DEGs were mainly down-regulated, suggesting that the viral suppressor totally inhibited silencing response. We detected just a few DEGs of JA-dependent signaling, which indicates a minor role of this pathway in the defense response of cucumber plants to CGMMV infection. The same applies to the expression of PR proteins in response to CGMMV infection. Finally, the TFs belonging to the WRKY and NAC families were the most sensitive to the virus inoculation among all TFs that had changed their expression.

## Figures and Tables

**Figure 1 life-11-01064-f001:**
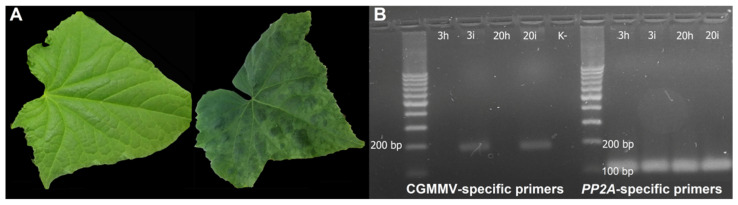
Symptoms of CGMMV infection and its confirmation by RT-PCR. (**A**) A cucumber leaf from the infected plant at 3 dpi (**left**) and 20 dpi (**right**); (**B**) RT-PCR analysis with CGMMV-specific primers and control *PP2A* housekeeping gene primers. 3h, control healthy plants at 3 dpi; 3i, plants inoculated with CGMMV at 3 dpi; 20h, control healthy plants at 20 dpi; 20i, plants inoculated with CGMMV at 20 dpi.

**Figure 2 life-11-01064-f002:**
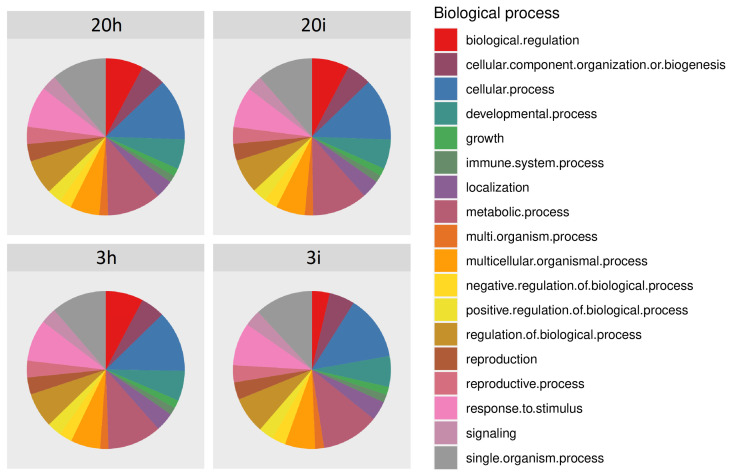
Distribution of transcripts by Gene Ontology (GO) terms for biological process.

**Figure 3 life-11-01064-f003:**
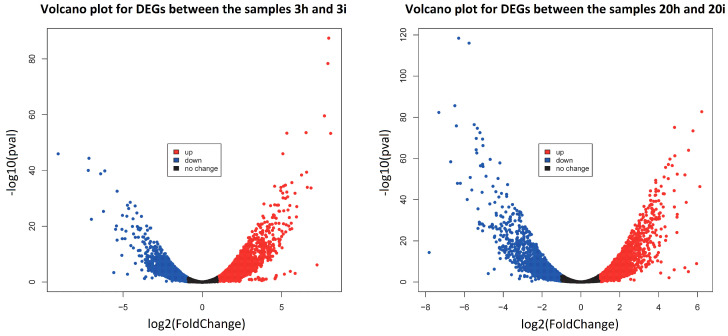
Volcano plot for differently expressed genes between the pairs of healthy-infected samples at 3 dpi (left) and 20 dpi (right). Up-regulated genes are shown in red, down-regulated genes are shown in blue, and the genes without significant expression change are shown in black.

**Figure 4 life-11-01064-f004:**
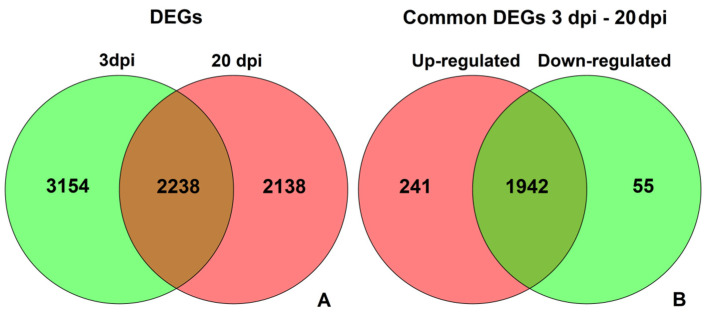
Venn diagrams showing the numbers of (**A**) common and unique DEGs between 3 dpi and 20 dpi and (**B**) common DEGs at both 3 dpi and 20 dpi that were up-regulated and down-regulated in infected plants at both time points or had different expression patterns.

**Figure 5 life-11-01064-f005:**
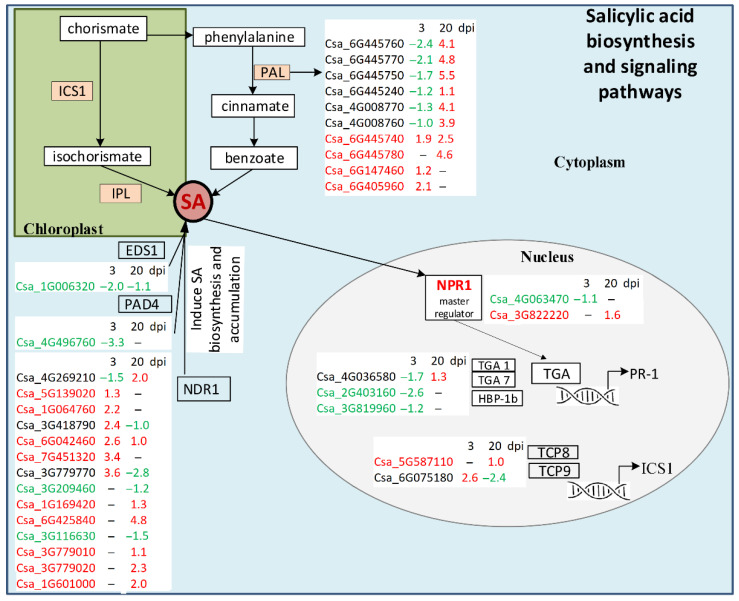
Schematic representation of SA biosynthesis and signaling pathways. See the main text for details. DEGs found in infected cucumber plants in comparison with healthy control plants are shown in white boxes. DEGs that were up-regulated only are shown in red, down-regulated DEGs are given in green, and DEGs that were both up- and down-regulated are shown in black.

**Figure 6 life-11-01064-f006:**
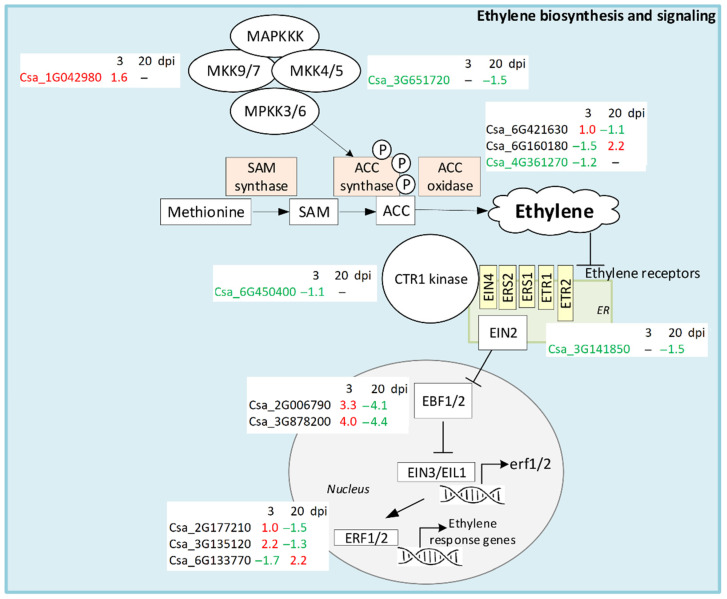
Schematic representation of ET biosynthesis and signaling pathways. DEGs found in infected cucumber plants in comparison with healthy control plants are shown in white boxes. DEGs that were up-regulated only are shown in red, down-regulated DEGs are given in green, and DEGs that were both up- and down-regulated are shown in black.

**Figure 7 life-11-01064-f007:**
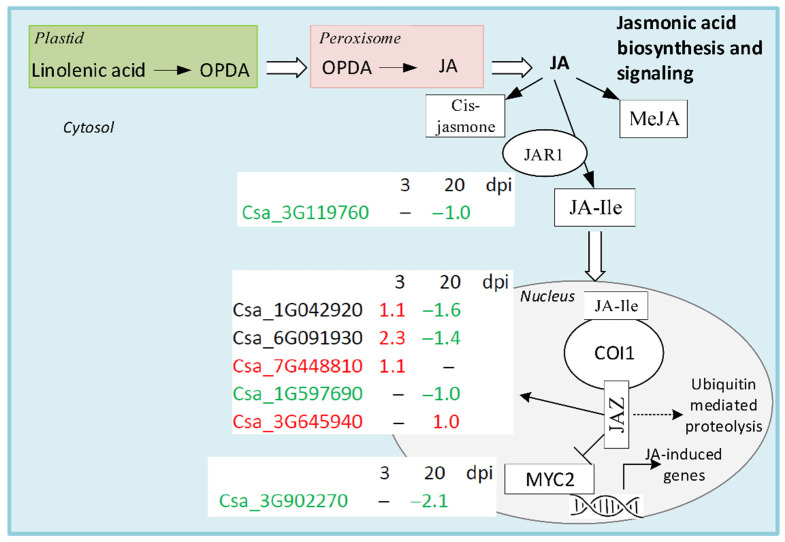
Schematic representation of JA biosynthesis and signaling pathways. DEGs found in infected cucumber plants in comparison with healthy control plants are shown in white boxes. DEGs that were up-regulated only are shown in red, down-regulated DEGs are given in green, and DEGs that were both up- and down-regulated are shown in black.

**Figure 8 life-11-01064-f008:**
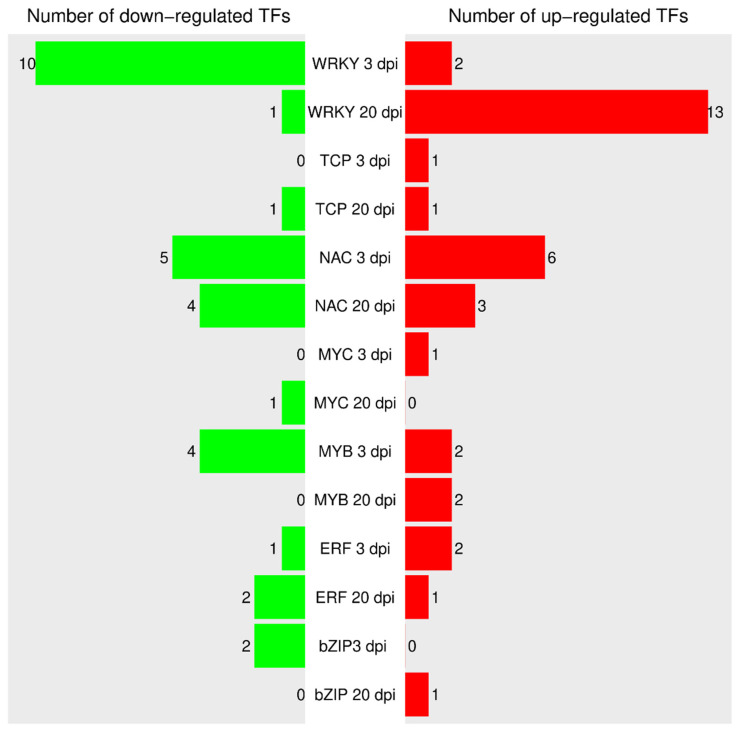
Up- or down-regulation of different DEGs encoding TFs involved in response to CGMMV infection.

**Figure 9 life-11-01064-f009:**
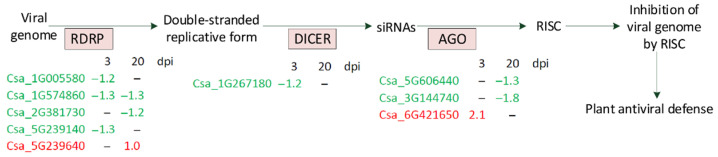
Schematic representation of DEGs encoding different members of VIGS in response to CGMMV.

**Table 1 life-11-01064-t001:** Statistics of transcriptome sequencing.

Transcriptome id	Days Post Inoculation	Number of Reads	Number of Transcripts	Median Transcript Length
3h	3	94,646,812	51,956	800
3i	3	100,708,366	54,781	846

**Table 2 life-11-01064-t002:** Top 10 genes with the greatest expression change in response to CGMMV infection.

Gene ID	log2(FoldChange)	Gene Annotation
**3 dpi**		
Csa_3G105950	−9.1	MTP-3 metal tolerance protein type 3-like
Csa_6G191570	8.1	Histone H4
Csa_2G030610	8	Probable histone H2A.5
Csa_4G290220	7.9	Histone H3.2
Csa_1G612925	7.7	Histone H3.2
Csa_2G429030	7.2	Benzylalcohol O-benzoyl transferase
Csa_2G176690	−7.2	PRP1 proline-rich protein
Csa_4G001950	−7.1	Sucrose synthase 2-like
Csa_6G084600	−7	Uncharacterized
Csa_7G064050	6.9	Uncharacterized
**20 dpi**		
Csa_6G448740	−7.8	Threonine dehydratase biosynthetic, chloroplastic
Csa_4G154320	−7.3	Polygalacturonase-inhibiting protein
Csa_2G429030	−6.7	Benzylalcohol O-benzoyltransferase
Csa_3G778180	−6.5	Uncharacterized
Csa_4G308490	−6.4	Benzylalcohol O-benzoyltransferase
Csa_4G000820	−6.3	Uncharacterized
Csa_1G050360	−6.3	Malate synthase, glyoxysomal-like
Csa_2G176190	6.2	Repetitive proline-rich cell wall protein 3-like
Csa_3G076580	−6.2	Plastid lipid- associated protein
Csa_2G287040	6.1	Uncharacterized

## Data Availability

Raw sequence reads for all samples were deposited to NCBI SRA archive (https://www.ncbi.nlm.nih.gov/sra) under project number PRJNA646644, SRS7015510–SRS7015513.

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
