# Peer review of "Transcriptomic Analysis of Genes Involved in Plant Defense Response to the Cucumber Green Mottle Mosaic Virus Infection"

_life, 2021, doi:10.3390/life11101064_

Round 1
Reviewer 1 Report
Remarks
This is a well executed work. The authors in this work probed into a interesting issue, i.e., what kind of signalling pathways/defence responses are induced/suppressed in early and late infection stages of cucumber green mottle mosaic virus (CGMMV) in Cucumis sativus. Importantly, the authors identified that salicylic acid and ethylene-signalling pathways were influenced differently during early and late stages of CGMMV infection. However, in both infection stages RNA silencing, one of the principal defence mechanisms against viruses were suppressed. Finally, Transcription factors WRKY and NAC were found to be highly responsive to CGMMV infection.
The work is overall very good however, the quality of writing is a big concern. In my opinion, a thorough modification in text and inclusion/removal of a few results will justify its publication.
Experimental-
- Result 3.1- From Fig.1 B it seems that the virus level is quite same in 3i and 20i plants. Are these systemically infected leaves? The authors should include a sentence to explain this since the difference in symptoms are quite significant.
- Result 3.2 Authors should include a sentence in the text explaining why ‘Immune system process’ and ‘Response to stimulus’ received primary attention in further DEGs studies (line 185-186)?
- Since there is a shift towards downregulated genes in 20i samples, I wonder if the there is any common downregulated DEGs between 3dpi and 20dpi (similar to Fig 4B). Authors can emphasise this by including it in Fig 4 as a Venn diagram, or by including it as a supplementary table.
- Line 238- Which data is referred to when authors are saying ‘more than 100 times at 20 dpi’?
- Result 3.5- Authors should include a few sentences to explain the basis of selecting these 9 genes amongst DEGs
Writing-
- Lines 283-387 is complete repetition of lines 157-282.
- Rephrase lines 215-217
- Rephrase lines 237-240
- Include reference number for lines 243-245
- Rephrase lines 394-395
- The discussion is extremely text heavy and it should be reduced to atleast half of its current size. Many parts of the text are quite well known information; the authors could cite literatures, rather than explaining the whole thing in the text. Also, certain things can be eliminated, for say, since JA-dependent defense response is not that important in CGMMV infection, it can be eliminated from the main text. If the authors wish they can keep it in some form in the supplementary materials. Same goes for PR proteins.
- Also, the discussions are mostly from the perspective of plants biology or interactions with pathogens other than viruses. In my opinion, it would be more informative if the authors could include some examples to correlate of the findings with respect to virus infection (CGMMV or other related viruses), if there are reports available, (for example, differential regulation of ethylene biosynthesis due to CGMMV was reported in DOI: 10.1038/s41598-017-17140-4).
Author Response
We are grateful to the Reviewer for thorough analysis of our Manuscript and his/her valuable comments. Our answers are given below.
Experimental-
- Result 3.1- From Fig.1 B it seems that the virus level is quite same in 3i and 20i plants. Are these systemically infected leaves? The authors should include a sentence to explain this since the difference in symptoms are quite significant.
For the experiments, we took infected leaves of the upper tier, located above the inoculated ones and infected due to systemic spread of the virus. To confirm the presence of CGMMV in the leaf samples, we performed qualitative PCR analysis. The results verified the presence of viral RNA in both samples. As it is known for CGMMV, the mosaic symptoms appear on the plant not earlier than 7 dpi. This explains the different phenotype of leaves at 3 dpi and 20 dpi (Figure 1). The explanation was added to the text (Lines 114-116, 173-174).
- Result 3.2 Authors should include a sentence in the text explaining why ‘Immune system process’ and ‘Response to stimulus’ received primary attention in further DEGs studies (line 185-186)?
The aim of this work was to study genes involved in defense response of cucumber plants to the pathogenic CGMMV strain (stimulus), therefore, ‘Immune system process’ and ‘Response to stimulus’ received primary attention in further DEGs studies. This explanation was added to the Results section (Lines 198-199).
- Since there is a shift towards downregulated genes in 20i samples, I wonder if the there is any common downregulated DEGs between 3dpi and 20dpi (similar to Fig 4B). Authors can emphasise this by including it in Fig 4 as a Venn diagram, or by including it as a supplementary table.
Indeed!! There are 55 common down-regulated DEGs between 3 dpi and 20 dpi, as we can see in Figure 4B. We changed the heading of Figure 4B from “infected-up and healthy-up” to “up-regulated and down-regulated”. We believe that this correction will make the meaning clearer.
- Line 238- Which data is referred to when authors are saying ‘more than 100 times at 20 dpi’?
We refer to our data presented in Table 2. There is a typo in the text, which we corrected: not ‘down-regulated more than 100 times at 20 dpi’, but ‘down-regulated at 20 dpi (log2(FoldChange) = -6.7)’. (Line 251).
- Result 3.5- Authors should include a few sentences to explain the basis of selecting these 9 genes amongst DEGs
Nine DEGs were selected for verification of changes in gene expression levels revealed by RNA-seq based on the following considerations. The selected genes belong to different functional groups: genes encoding proteins involved in signaling pathways, TF genes induced by biotic stress, and genes involved in RNA-silencing; their minimum expression level according to transcriptome analysis data should be at least 50 counts.
The explanation was added to the Results section (Lines 327-333)
Writing-
- Lines 283-387 is complete repetition of lines 157-282.
We deleted the repetition.
- Rephrase lines 215-217
Lines were rephrased. (Lines 230-231)
- Rephrase lines 237-240
Lines were rephrased. (Lines 250-254)
- Include reference number for lines 243-245
The reference number was added.( Line 257)
- Rephrase lines 394-395
Lines were rephrased. (Lines 301-302)
- The discussion is extremely text heavy and it should be reduced to atleast half of its current size. Many parts of the text are quite well known information; the authors could cite literatures, rather than explaining the whole thing in the text. Also, certain things can be eliminated, for say, since JA-dependent defense response is not that important in CGMMV infection, it can be eliminated from the main text. If the authors wish they can keep it in some form in the supplementary materials. Same goes for PR proteins.
The goal of our work was to study genes involved in defense response of cucumber plants to pathogen with a special focus on SA, ET and JA signaling, VIGS genes, as well as genes encoding PR-proteins and TFs, so all the revealed genes were provided only with the necessary information concerning their functions and their role in plant defense. So that the interested reader could immediately find the necessary information.
We decided to include the description of JA signaling pathway and PR-proteins in the Discussion section because a priori we did not know whether the infection affects the expression of the genes involved.
- Also, the discussions are mostly from the perspective of plants biology or interactions with pathogens other than viruses. In my opinion, it would be more informative if the authors could include some examples to correlate of the findings with respect to virus infection (CGMMV or other related viruses), if there are reports available, (for example, differential regulation of ethylene biosynthesis due to CGMMV was reported in DOI: 10.1038/s41598-017-17140-4).
We added this to the Discussion section (Lines 478-485).
Reviewer 2 Report
The manuscript is an excellent study exploring the response of cucumber plants to one of the global threat pathogens, cucumber green mottle mosaic virus (CGMMV) within cucurbits industry, which causes severe symptoms on leaves and fruits damage. Could the authors speculate more on the application of RNA-Seq in viruses studies such as differential gene expression.
The authors should correct the grammatical errors in the manuscript.
Author Response
We are grateful to the Reviewer for thorough analysis of our Manuscript and his/her valuable comments. Our answers are given below.
The manuscript is an excellent study exploring the response of cucumber plants to one of the global threat pathogens, cucumber green mottle mosaic virus (CGMMV) within cucurbits industry, which causes severe symptoms on leaves and fruits damage. Could the authors speculate more on the application of RNA-Seq in viruses studies such as differe ntial gene expression.
We added some speculations on this topic to the Introduction and Discussion sections (Lines 84-94, 478-485)
The authors should correct the grammatical errors in the manuscript.
The grammatical errors were corrected.
Reviewer 3 Report
The manuscript entitled "Transcriptomic Analysis of Genes Involved in Plant Defense Response to the Cucumber Green Mottle Mosaic Virus Infection" is a nice study and well written. However, it needs the following minor changes before acceptance.
Comments 1# In every figure there are so many IDs are mentioned. Add their gene name (Short name only), which will help to understand the reader more.
Comments 2# In figure 5, the font size of the text in the image is not uniform; the authors need to redraw it again.
Comments 3# In figure 1B, mention the size of the lower band also.
Minor Comments#
Line 83- Authors need to put a full form of C. sativus as it is the first time used.
Line 201- Remove the word "see" (see Table S3)
Line 155- PP2A housekeeping gene; line 698---GbNAC1 (In line 699,697 etc. ); gene name should be italic throughout the manuscript.
Line- 359, 360, 746, 747, 748; Try to avoid abbreviations such as Ile TD AGO, RISC, PTGS which are only used one time.
Line 4- Salicylic acid (SA) is already mentioned in line 50.
Author Response
We are grateful to the Reviewer for thorough analysis of our Manuscript and his/her valuable comments. Our answers are given below.
Comments 1# In every figure there are so many IDs are mentioned. Add their gene name (Short name only), which will help to understand the reader more.
Gene IDs with the names of the proteins encoded by the genes are given in the text of the manuscript. Some proteins are encoded by polymorphous genes, for example, we found 10 differently expressed genes encoding PAL (Figure 5). The short names of proteins encoded by the genes are shown in the figures in black frames. We believe this information is necessary and sufficient for a better understanding.
Comments 2# In figure 5, the font size of the text in the image is not uniform; the authors need to redraw it again.
The figure was corrected.
Comments 3# In figure 1B, mention the size of the lower band also.
It was added.
Minor Comments#
Line 83- Authors need to put a full form of C. sativus as it is the first time used.
Done. Full name was added. (Line 95)
Line 201- Remove the word "see" (see Table S3)
Done. (Line 215)
Line 155- PP2A housekeeping gene; line 698---GbNAC1 (In line 699,697 etc. ); gene name should be italic throughout the manuscript.
It was corrected.
Line- 359, 360, 746, 747, 748; Try to avoid abbreviations such as Ile TD AGO, RISC, PTGS which are only used one time.
It was corrected.
Line 4- Salicylic acid (SA) is already mentioned in line 50.
It was corrected.